# Biophysical Changes of Leukocyte Activation (and NETosis) in the Cellular Host Response to Sepsis

**DOI:** 10.3390/diagnostics13081435

**Published:** 2023-04-16

**Authors:** Matt G. Sorrells, Yurim Seo, Melia Magnen, Bliss Broussard, Roya Sheybani, Ajay M. Shah, Hollis R. O’Neal, Henry T. K. Tse, Mark R. Looney, Dino Di Carlo

**Affiliations:** 1Cytovale Inc., San Francisco, CA 94134, USA; 2Departments of Medicine and Laboratory Medicine, University of California San Francisco, San Francisco, CA 94143, USA; 3Department of Biological Sciences, Nicholls State University, Thibodaux, LA 70310, USA; 4LSU Health Sciences Center/Our Lady of the Lake Regional Medical Center, Baton Rouge, LA 70808, USA; 5Departments of Bioengineering and Mechanical and Aerospace Engineering, University of California Los Angeles, Los Angeles, CA 90095, USA

**Keywords:** sepsis, neutrophil extracellular traps, sepsis diagnostics, neutrophil, monocyte, novel biomarkers

## Abstract

Sepsis, the leading cause of mortality in hospitals, currently lacks effective early diagnostics. A new cellular host response test, the IntelliSep test, may provide an indicator of the immune dysregulation characterizing sepsis. The objective of this study was to examine the correlation between the measurements performed using this test and biological markers and processes associated with sepsis. Phorbol myristate acetate (PMA), an agonist of neutrophils known to induce neutrophil extracellular trap (NET) formation, was added to whole blood of healthy volunteers at concentrations of 0, 200, and 400 nM and then evaluated using the IntelliSep test. Separately, plasma from a cohort of subjects was segregated into Control and Diseased populations and tested for levels of NET components (citrullinated histone (cit-H3) DNA and neutrophil elastase (NE) DNA) using customized ELISA assays and correlated with ISI scores from the same patient samples. Significant increases in IntelliSep Index (ISI) scores were observed with increasing concentrations of PMA in healthy blood (0 and 200: *p* < 10^−10^; 0 and 400: *p* < 10^−10^). Linear correlation was observed between the ISI and quantities of NE DNA and Cit-H3 DNA in patient samples. Together these experiments demonstrate that the IntelliSep test is associated with the biological processes of leukocyte activation and NETosis and may indicate changes consistent with sepsis.

## 1. Introduction

Sepsis is a leading cause of morbidity and mortality in hospitals worldwide, with substantial unmet need for the rapid diagnosis and delivery of precision therapies. Early intervention in sepsis is essential in reducing morbidity and mortality. As such, sepsis management guidelines recommend rapid clinical recognition of the disease followed by interventions to optimize patient outcomes [1]. However, most cases of sepsis present to the emergency department (ED), where ED physicians must take action often before adequate, objective diagnostic and prognostic data are available [2]. A new class of tests was recently cleared by the U.S. Food and Drug Administration (FDA) for early detection of sepsis based on physical biomarkers of immune cells [3], which can be rapidly assessed in the ED, addressing this gap. Here, we link the results of this new cellular host response test with other molecular markers associated with immune cell activation and sepsis, such as formation of neutrophil extracellular traps (NETs) and the systemic signature of NETs in the bloodstream.

The consensus definition of sepsis has evolved over the last three decades [4]. The current Sepsis-3 designation, adopted in 2016 by the Third International Consensus Task Force, defines the syndrome as “life-threatening organ dysfunction caused by a dysregulated host response to infection” [5]. Briefly, this timeline of events begins with a host infection leading to a sustained immune response that becomes systemic following a proinflammatory cascade [6,7,8]. Though sustained immune activation localized to the region of infection aids in fighting the infection, sustained systemic activation can result in the injury to other tissues [6,7,8]. Ultimately, this leads to organ dysfunction/failure, shock, and death if the process runs its complete course [6,7,8]. Currently available methods of assessing infection and organ dysfunction cannot differentiate between chronic and acute organ dysfunction, assess whether the organ dysfunction has an explanation other than infection, or attribute dysfunction specifically to a dysregulated host immune response [9]. Therefore, there is a need for an objective biological tool to assess dysregulated immunity early in the ED presentation to aid in timely sepsis diagnosis and appropriate intervention.

Several approaches have been explored for evaluating activation of the immune system. These include structural feature analysis through light microscopy [10,11], quantification of cell surface markers through immunofluorescent flow cytometry [12,13,14,15], and even some promising efforts in correlating transcriptomic information with innate immune activation [16,17]. While providing considerable value, these methods suffer from a slow turnaround time and the need for specialized expertise, which makes them ill-suited for use in an emergency setting or where low complexity and affordable options are needed for repeated use over time or for large subject populations.

During innate immune activation, neutrophil and monocyte membrane receptors sense pathogen-associated molecular pattern molecules (PAMPs) and respond via chemotaxis, phagocytosis, cytokine signaling, generation of reactive oxygen species (ROS), release of microbicidal granular contents, and in some cases, ultimately, the expulsion of NETs into the extracellular space (Figure 1) [18,19,20]. NETs consist of scaffolded chromatin complexed with an array of citrullinated histones (CitH3) and granular proteins, which physically capture and kill or attenuate invading microbes [19]. Although NETs play an important role in the host antimicrobial defense strategy, their hyperproduction or improper localization due to dysregulated immunity is known to exert histotoxic and hypercoagulative effects in vivo, contributing to both chronic and acute inflammation and exacerbating septic injury [21,22,23]. Studies have reported elevated levels of NET-associated histones (H3) in clinical samples of septic patients and in lipopolysaccharide (LPS) induced septic mice, while studies using non-human primate models of sepsis further linked H3 to acute renal failure [24,25]. Additionally, higher plasma levels of circular free DNA derived from NETs (cf-DNA/NETs) correlated with increased risk of secondary inflammation and sepsis in a clinical pilot study of ICU patients with traumatic injury [26].

Early stages of NETosis, prior to expulsion, involve a dramatic sequence of intracellular restructuring, including morphogenesis of the nucleus, which is known to be the largest, most structurally integral, and stiffest cellular organelle in leukocytes [27]. The many molecular nuclear and cytoskeletal modifications that occur during activation and NETosis temporally alter the biophysical profiles of leukocytes, which, if assayed rapidly, may allow an evaluation of immune activation in real time, potentially providing visibility to early stages of immune dysfunction prior to severe clinical manifestation of multi-organ dysfunction [28]. Measuring these biophysical changes could be diagnostically beneficial to better guide clinical care.

The IntelliSep test is the first FDA-cleared cellular host response in vitro diagnostic test for sepsis that measures the biophysical changes of immune cells. This test leverages microfluidics in combination with high-speed imaging and a proprietary algorithm to quantify biophysical characteristics of thousands of leukocytes per test from a 100 µL sample of routinely collected whole blood [29,30]. The test yields an IntelliSep Index (ISI) in under ten minutes, a score between 0.1–10.0 that is stratified into three discrete interpretation bands based on probability of sepsis: Band 1, Band 2, and Band 3 [30]. The diagnostic performance of the IntelliSep test when applied to patients presenting to the ED with signs or suspicion of infection was previously evaluated in several clinical studies [30,31,32].

We hypothesize that the immunological changes to neutrophils and monocytes during activation, including those resulting from nuclear rearrangements during early stages of NETosis, result in biophysical changes measurable by the IntelliSep test, and that both may predict the probability of sepsis in patient samples. This present study consists of two experiments that evaluate this hypothesis. First, an experiment was designed to measure the biophysical properties (through ISI) of leukocytes from whole blood activated in vitro with phorbol myristate acetate (PMA), an agent known to induce NETosis [33,34]. Secondly, to understand possible correlations between the ISI and NET concentrations in plasma of patients suspected of infection, the presence of NETs was quantified by ELISA [35] and compared to ISI from a cohort of patients with varying ISI levels [30,31].

## 2. Materials and Methods

### 2.1. In Vitro Experiments

#### 2.1.1. Reagents and Solutions

Phorbol myristate acetate (PMA, Sigma Aldrich, St. Louis, MO, USA), which induces NETosis by bypassing membrane receptor-PAMP interactions with direct stimulation of Protein Kinase C (PKC) [33,34], was prepared as a 1 mM stock solution in dimethyl sulfoxide (DMSO, Fisher BioReagents, Pittsburgh, PA, USA), aliquoted, and stored at −20 °C to minimize freeze-thaw cycles. For each day the PMA Assays were conducted, an aliquot of 1 mM was defrosted and diluted in DMSO to 20 µM and 10 µM for the preparation of spiked blood samples.

##### 2.1.2. PMA Assays

Healthy blood samples were provided by Our Lady of the Lake Blood Donor Center, Baton Rouge, LA, from April to August 2022, from 18 donors who completed a questionnaire indicating no signs or symptoms of current illnesses. The study was approved by WCG Institutional Review Board (Study #1303991; WCG IRB #1-1407262-1) and the study team obtained written informed consent from all participants. Aliquots of EDTA-anticoagulated whole blood were treated with PMA to a final concentration of either 200 nM or 400 nM, or with an equal volume of 1X PBS (Gibco), and incubated in a water bath at 37 °C. After 10 min, the PMA- or PBS-spiked blood samples were removed from the water bath and 100 µL volumes were used to perform the IntelliSep Test (Cytovale, San Francisco, CA, USA) [33]. All IntelliSep tests were performed within six hours of venipuncture for healthy blood sample collection to avoid possible sample degradation.

### 2.2. Clinical Studies

#### Study Population

Subjects were enrolled in three similar but distinct prospective cohort studies between February 2016 and September 2019 at two academic medical centers, namely Our Lady of the Lake Regional Medical Center and Baton Rouge General Medical Center, in Baton Rouge, LA [30,31]. The studies were approved by the Louisiana State University Health Sciences Center Institutional Review Board (IRB) as well as by local, hospital-specific IRBs as appropriate (study 1: Louisiana State University Health Sciences Center—New Orleans, Human Subjects Research Protection Program and Institutional Review Board: LSUHSC-NO #8964; study 2: Louisiana State University Health Sciences Center—New Orleans, Human Subjects Research Protection Program and Institutional Review Board: LSUHSC-NO #9749; study 3: Louisiana State University Health Sciences Center—New Orleans, Human Subjects Research Protection Program and Institutional Review Board: LSUHSC-NO # 19-019, Franciscan Missionaries of Our Lady Institutional Review Board: FRANU # 2019-012, and Baton Rouge General Institutional Review Board: number 2018-017). Research personnel actively screened for subjects using the electronic health record and obtained informed consent from participants in accordance with IRB-approved protocols.

Participants presented to the ED with signs or suspicion of infection. Signs of infection were defined as having two or more systemic inflammatory response syndrome (SIRS) criteria [4] with at least one being aberration of body temperature or white blood cell (WBC) count. Suspicion of infection was defined as a culture of bodily fluid being ordered by a clinician. Subjects were excluded for the following conditions: expected palliative care, history of hematologic disorders, receipt of cytotoxic chemotherapy within 3 months of arriving at the ED, prisoners, transfers from other acute care facilities, and those unwilling or unable to consent. Subjects were followed by retrospective chart review for outcome information and determination of disease status was performed by a retrospective physician adjudication. All study personnel were blinded to the results of the ISI.

Enrolled participants were asked to provide additional consent for remnant specimen storage. Plasma samples were collected and stored for this subset of subjects, 147 of which were identified for analysis in this study.

An additional down-sampling was performed on the enrolled population as a part of this study, in which 81 subjects were selected for being Control or Diseased. Control subjects were defined as those who were adjudicated as not infected, whose Sequential Organ Failure Assessment (SOFA) score on the day of enrollment/ED presentation was less than two, and who had a total hospital length of stay of less than 3 days. Diseased subjects were defined as those who were adjudicated as septic and had a maximum SOFA score over 3 days occurring on the day of enrollment/ED presentation.

### 2.3. IntelliSep Test and Results

After obtaining consent, research coordinators collected peripheral whole-blood samples for the IntelliSep test. As described in O’Neal et al. [31], the IntelliSep test (Cytovale, San Francisco, CA) uses 100 µL of whole blood to assess mechanical and morphological changes that occur during leukocyte activation in less than 10 min. The ISI, the test output, is a single score ranging from 0.1 to 10.0 that is segmented into three interpretation bands based on the probability of sepsis: Band 1, Band 2, and Band 3. The remnant blood was processed, and plasma from the remnant of these samples was stored at −80 °C. Procalcitonin (PCT) concentration measurements were performed via Quantigen Genomic Services (Fishers, IN, USA). PCT is a non-specific clinical biomarker that is sometimes used for the detection of systemic bacterial infection and sepsis [36].

The development and validation of the IntelliSep test’s underlying diagnostic algorithm are detailed in Guillou et al. [30]; as part of the image analysis, WBC subpopulations (lymphocytes, neutrophils, and monocytes) are identified and enumerated by measuring cell size and optical intensity, using an automated clustering algorithm. The clustering algorithm was validated by comparing the cell clusters found by the automated clustering to the fluorescent data obtained from PE-Cy5 CD45 and PE-CD66B stains using conventional flow cytometry techniques. One of the metrics underlying the IntelliSep test’s diagnostic algorithm is visco-elastic inertial response (VEIR), calculated for the neutrophil and monocyte subpopulations. VEIR is a measure of cell oscillations when interacting with the extensional flow within the microfluidic junction of the IntelliSep cartridge, and is dependent on cell density, size, and visco-elastic properties [30].

### 2.4. Quantification of NET Content

As a measure of NETs, neutrophil elastase (NE)-DNA and citrullinated histone H3(Cit-H3)-DNA complexes were quantified in patient plasma samples using previously reported custom ELISAs [35,37]. Capture antibodies for anti-neutrophil elastase (Santa Cruz Biotechnology, SC-55549) and anti-citrullinated histone H3 (Abcam, ab-5103) were coated on Maxisorp wells (Nunc) followed by the anti-DNA-HRP detection antibody (Roche, Cell Death Detection ELISA kit).

### 2.5. Statistical Analysis

*p*-values were computed from an unpaired two-sample Welch’s *t*-test, unless stated otherwise. An alpha value of 0.05 was used unless stated otherwise. Two-sided linear least-square regressions are presented with Pearson correlation coefficients (*r*-values).

## 3. Results

### 3.1. Activation of Neutrophil and Monocytes through PMA Is Detected through IntelliSep Measurements

Neutrophils and monocytes stimulated with PMA showed greater visual deformation than unstimulated ones, with increasing deformation at higher PMA concentrations (Figure 2A). Significant increases in IntelliSep Index scores (ISI) were observed as PMA concentration was increased from 0 to 400 nM (Figure 2B). This increase in ISI score was observed to be primarily due to increases in VEIR of neutrophils and monocytes. Likewise, as the PMA concentration in the blood was increased, an increasing number of these samples were given Band 3 (high probability of sepsis) scores using the IntelliSep test (Figure 2B). As expected [38], at higher PMA concentrations, the concentration of cells in the samples was observed to decline (data not shown).

### 3.2. Quantity of NETs in Patient Samples Is Correlated with Sepsis and IntelliSep Index Measurements

To investigate the relationships between NET formation, sepsis, and the IntelliSep measurements, neutrophil-elastase DNA (NE-DNA) and citrullinated H3 DNA (Cit-H3 DNA) were measured [35,38]. When applying selection criteria to further segregate Control and Diseased subjects as the test group (n = 81, Table 1), both markers of NETs in the sample were observed to correlate significantly with both disease status and IntelliSep Index. Significant 1.48-fold and 2.56-fold increases in Cit-H3 (0.48 Band 1, 0.71 Band 3, *p* < 0.05) and NE DNA quantity (0.25 Band 1, 0.65 Band 3, *p* < 0.05), respectively, were observed in the Diseased group compared to the Control group (Figure 3). When comparing IntelliSep measurements and markers of NETs quantity, we noted modest correlation coefficients of 0.27 and 0.31 between ISI values and Cit-H3 and NE DNA quantity, respectively (Figure 4A). Neutrophil VEIR measurements showed lower degrees of correlation with Cit-H3 and NE DNA quantities with coefficients of 0.124 and 0.233, respectively (Figure 4B). When comparing NET biomarkers from the blood of Band 3 patients to those from Band 1 patients, we observed significant 2- and 3-fold increases in Cit-H3 (0.43 Band 1, 0.86 Band 6, *p* < 0.01) and NE-DNA (0.25 Band 1, 0.72 Band 3, *p* < 0.001) quantities, respectively (Figure 4C). Finally, PCT concentration levels of subjects were compared to ISI and Cit-H3 DNA quantities. ISI yielded higher Pearson correlation coefficients with Cit-H3 DNA (0.306) and NE DNA (0.271) quantities than PCT (0.12 for Cit-H3 DNA and 0.208 for NE DNA). Additionally, Cit-H3 DNA and NE-DNA values fell along a more even distribution across the range of ISI values, versus a log-normal distribution across the range of PCT values, with ELISA values biased towards lower PCT levels (Figure 5).

## 4. Discussion

Previously, we have demonstrated the performance of the IntelliSep test and its corresponding ISI as an early diagnostic test for sepsis and in risk-stratifying patients with signs or suspicion of infection in the emergency department [30,32]. The findings in this study provide evidence to support a connection between the biophysical measurements observed in the IntelliSep test and biological phenomena occurring during leukocyte activation in a dysregulated host response to infection and sepsis. The results of this study appear to show that ISI increases when neutrophils are activated in vitro with PMA to form NETs as well as a positive correlation between the ISI and biomarkers associated with NETs in samples collected from patients adjudicated to have sepsis. This may reflect similar changes in leukocyte biophysics measured by the IntelliSep test in both in vitro and in vivo activation. We hypothesize that leukocyte biophysical changes result from nuclear membrane integrity loss and chromatin de-condensation during cell activation, preceding NETosis. In addition, notably, we observed the same biophysical changes in monocytes as in neutrophils, which is consistent with recent reports that monocytes might also be capable of releasing NET-like structures [20]. The same changes, however, would not likely be observed in lymphocytes according to previous findings [29], possibly reflecting the fundamental differences in mechanism between the innate and the adaptive immune systems.

Previously, an increase in NET formation had been documented in septic patients, and high concentrations of NETs had been shown to be associated with tissue damage. Lefrançais et al. observed an abundance of NETs in human subjects with acute respiratory distress syndrome (ARDS) from sepsis. Additionally, they observed that decreasing NETs through DNase treatment reduced lung injury and improved survival in murine models of severe bacterial pneumonia [35].

In this study, we observed that circulating NET fragments, quantifiable using two custom ELISAs, increase in patients adjudicated as septic and correlate with increasing ISI scores. Because the timeline for sepsis initiation in patients is difficult to elucidate and NETosis is believed to mark the very terminus of neutrophil activation, selective eligibility criteria were applied for the comparison of a Control group and a Diseased septic group. It is expected that those patients with significantly higher NET biomarker concentration in their peripheral blood sample have advanced disease, likely with manifesting multi-organ failure. Within these subpopulations, significant trends were observed in disease status, and NET biomarker concentration correlated with IntelliSep index and across its interpretation bands. NETs quantity increased steadily across the entire ISI range but did not increase across the range of PCT scores. Rather, NETs had a stronger correlation at higher PCT values and little correlation at lower ones and fell along a log-normal distribution across the range of PCT concentrations. This observation is consistent with other studies showing that PCT has a comparatively low sensitivity of 0.71 as an early sepsis diagnostic [39].

Although correlated with NET quantity, the IntelliSep test acts as a more sensitive early diagnostic of sepsis, since the biophysical changes detected by IntelliSep occur prior to overt NETosis, before significant NETs have formed and biochemical evidence of NETosis is present in the plasma [35]. Large scale NET formation in the bloodstream results in the NET-associated biomarkers we measured in this selected population.

Though these results support a link between the biophysical measurements assessed using the IntelliSep test and the state of neutrophil/monocyte activation and NETosis, the findings in this study do come with several limitations. Broadly, sepsis is a complex syndrome consisting of a multitude of biological pathways that all can affect the biophysical properties of neutrophils as well as other endpoints of neutrophil activation [6,8]. In this study, one source of activation, the PKC pathway activation via PMA stimulation, and one additional endpoint of activation, NETosis, were examined. Additionally, given that cell activation was stimulated with PMA, it is unclear what state of activation (oxidative stress, degranulation, nuclear rearrangement, and NETosis) the neutrophils achieved at each concentration of PMA. Since the sample preparation step of the IntelliSep test automatically washes away lysed cells, it is likely that the biophysical changes measured in these experiments do not include neutrophils and monocytes that have completed NETosis, whose cell membranes have been completely compromised. Though outside of the scope of this paper, one could more precisely determine the point of activation along this timeline for each concentration by performing a panel of measurements relevant to this timeline as done in Remijsen et al. [40].

Together these experiments support the biological underpinnings of the IntelliSep test as a rapid in vitro diagnostic test that could provide a window into a patient’s state of dysregulated immunity and could aid ED physicians in timely diagnosis of sepsis. As the ISI algorithm is trained on clinically adjudicated septic/not-septic subjects, it is likely to encompass signatures from a variety of pathways which may include the cellular processes of NETosis.

## Figures and Tables

**Figure 1 diagnostics-13-01435-f001:**
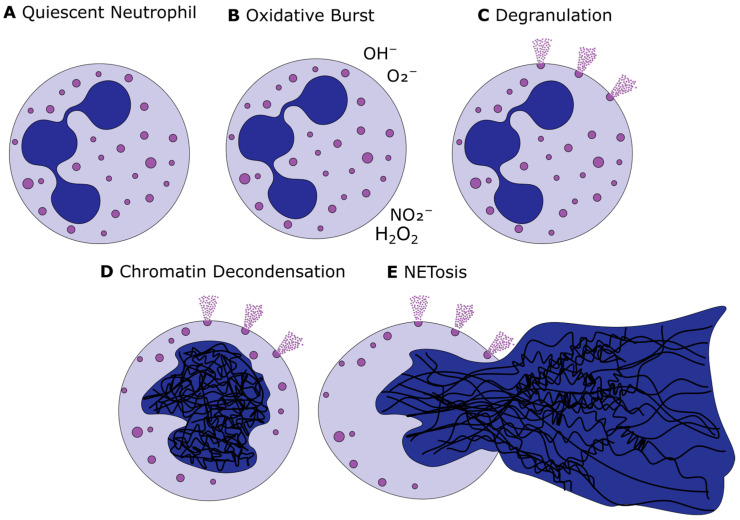
Diagram of neutrophil activation leading to neutrophil extracellular traps (NET)osis. Quiescent neutrophils (**A**) are stimulated by an agonist, resulting in the formulation of reactive oxygen species, known as oxidative burst, that aid in host defense (**B**). Neutrophils undergo degranulation, resulting in the release of a variety of anti-microbial proteins from granules (**C**). As activation proceeds, the nuclear chromatin of neutrophils decondenses from packed lobes into loosely arranged DNA (**D**), and the neutrophil cell membrane eventually ruptures, releasing the DNA as NETs into the extracellular environment (**E**).

**Figure 2 diagnostics-13-01435-f002:**
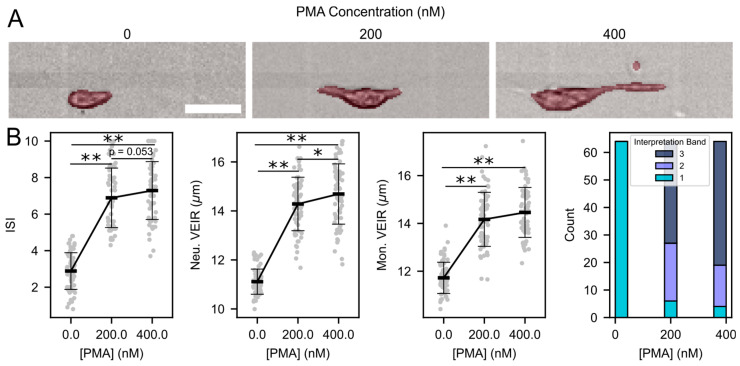
Effect of phorbol myristate acetate (PMA) on neutrophil deformability and IntelliSep measurements. Whole blood was incubated with PMA for 10 min then run through the IntelliSep test. (**A**) Representative images of neutrophils passing through microfluidic junction of the IntelliSep test. Neutrophils are pseudo-colored in maroon. Representative images were selected by identifying runs that yielded average IntelliSep Index (ISI) values for a given concentration and selecting neutrophil cell events that yielded average viscoelastic inertial response (VEIR) measurements for that run. Greater deformation of neutrophils was observed at increasing PMA concentration. Scale bar = 50 µm. (**B**) Aggregated data for IntelliSep measurements at varying PMA concentrations. Data sampling consisted of 18 donors and 3 repeats per donor per concentration of PMA. ISI and VEIR measurements were corrected for donor-to-donor variation by dividing by the donor average at 400 nM PMA concentration then rescaling the values to their original measurement range. We observed increases in ISI, cell oscillations (VEIR), and number of samples in Band 3 interpretation band with increasing PMA concentration. (* and ** denote *p* < 0.05 and *p* < 10^−10^ respectively; *p*-values computed with analysis of variance (ANOVA) and Tukey Honestly Significant Difference (HSD) post-hoc test).

**Figure 3 diagnostics-13-01435-f003:**
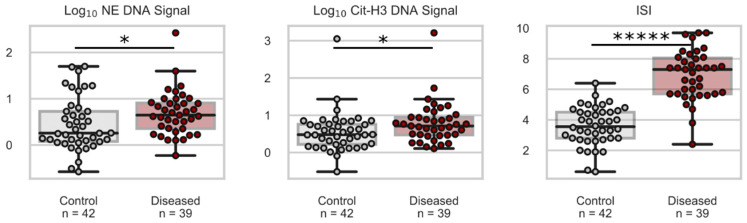
Quantity of neutrophil elastase (NE)-DNA, citrullinated histone H3 (Cit-H3)-DNA, and IntelliSep Index (ISI) for Control and Diseased patient plasma samples. * indicates *p* < 0.05 and ***** indicates *p* < 10^−5^.

**Figure 4 diagnostics-13-01435-f004:**
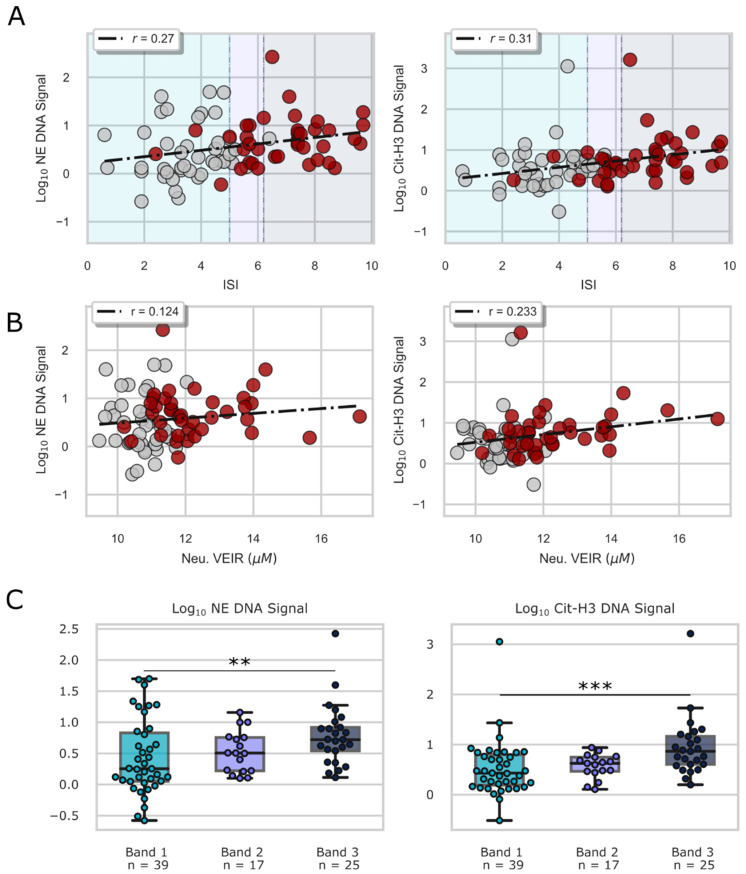
Quantity of neutrophil elastase (NE)-DNA and citrullinated histone H3 (Cit-H3)-DNA compared to IntelliSep Index (ISI). NE DNA and Cit-H3 DNA quantity vs. (**A**) IntelliSep Index and (**B**) neutrophil VEIR measurements. Gray and maroon markers indicate Control and Diseased patients, respectively. Teal, purple-blue, and dark-blue shaded regions indicate the range of ISI for the given interpretation band color. “r” values indicate Pearson correlation coefficients. (**C**) Quantity of NE DNA and Cit-H3 DNA across IntelliSep interpretation band scores. ** indicates *p* < 0.01, and *** indicates *p* < 0.001.

**Figure 5 diagnostics-13-01435-f005:**
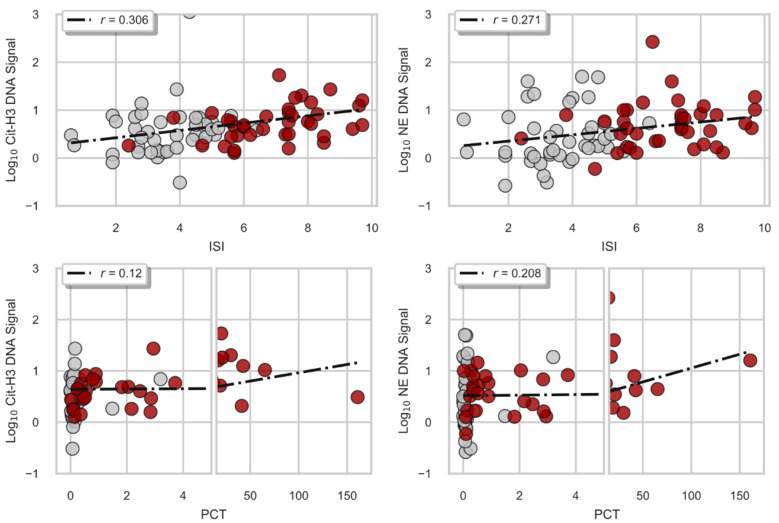
Correlation between neutrophil extracellular trap (NET) quantities, measured through citrullinated histone H3 (Cit-H3) DNA and neutrophil elastase (NE) DNA, and IntelliSep Index (ISI) or procalcitonin (PCT). “r” values indicate Pearson correlation coefficients. ISI yielded higher correlation coefficients with Cit-H3 DNA and NE DNA signals than PCT. Gray and maroon markers indicate Control and Diseased patients, respectively.

**Table 1 diagnostics-13-01435-t001:** Summary statistics on demographics and outcomes of subjects. *p*-values were computed using a two-sided *t*-test.

Category		Total N = 81	Control N = 42	Diseased N = 39	*p* Value
Age	Median (Q1–Q3)	58.00 (43.75–69.00)	52.50 (34.00–66.25)	62.00 (53.00–72.50)	*p* < 0.01
	Subjects ≥ 65 N (%)	28 (34.57)	11 (26.19)	17 (43.59)	ns
Biological Sex N (%)	Male	39 (48.15)	18 (42.86)	21 (53.85)	ns
	Female	42 (51.85)	24 (57.14)	18 (46.15)	ns
Race N (%)	Black	28 (34.57)	18 (42.86)	10 (25.64)	ns
	White	50 (61.73)	23 (54.76)	27 (69.23)	ns
	Other	3 (3.7)	1 (2.38)	2 (5.13)	ns
Comorbidities N (%)	Hypertension	18 (22.22)	11 (26.19)	7 (17.95)	ns
	Diabetes	7 (8.64)	4 (9.52)	3 (7.69)	ns
	Obesity	9 (11.11)	6 (14.29)	3 (7.69)	ns
	Cancer	4 (4.94)	2 (4.76)	2 (5.13)	ns
	Chronic Kidney Disease	3 (3.7)	2 (4.76)	1 (2.56)	ns
	Autoimmune Disease	3 (3.7)	2 (4.76)	1 (2.56)	ns
Infected by adjudication N (%)	Yes	39 (48.15)	0 (0.0)	39 (100.0)	*p* < 0.0001
Septic, by Sepsis-3 definition N (%)	Yes	39 (48.15)	0 (0.0)	39 (100.0)	*p* < 0.0001
All-Cause Cumulative In-Hospital Mortality N (%)	3-day	0 (0.0)	0 (0.0)	0 (0.0)	ns
	7-day	3 (3.7)	0 (0.0)	3 (7.69)	ns
	30-day	3 (3.7)	0 (0.0)	3 (7.69)	ns
Admitted to Hospital N (%)		57 (70.37)	19 (45.24)	38 (97.44)	*p* < 0.0001
Admitted to ICU N (%)		11 (13.58)	2 (4.76)	9 (23.08)	*p* < 0.05
SOFA, 3-day max (baseline subtracted) Median (Q1–Q3)		2.00 (0.00–4.00)	0.00 (0.00–1.00)	4.00 (3.00–6.00)	*p* < 0.0001
WBC (10^3^ cells/µL), Median (Q1–Q3)		13.16 (7.85–19.35)	9.65 (5.70–13.46)	17.30 (12.62–22.98)	*p* < 0.0001
Triage Temperature, Median (Q1–Q3)		98.30 (97.90–99.12)	98.20 (97.90–98.40)	98.90 (98.10–101.00)	*p* < 0.0001
Lactate Measured, N (%)		51 (62.96)	14 (33.33)	37 (94.87)	*p* < 0.0001
Lactate, Median (Q1–Q3)		2.10 (1.30–3.65)	1.80 (0.68–2.10)	2.90 (1.40–5.10)	*p* < 0.05
IntelliSep Index, Median (Q1–Q3)		5.00 (3.40–7.10)	3.55 (2.80–4.50)	7.30 (5.70–8.05)	*p* < 0.0001
Log_10_ Cit-H3 DNA Signal, Median (Q1–Q3)		0.61 (0.32–0.85)	0.48 (0.22–0.76)	0.71 (0.47–0.94)	*p* < 0.05
Log_10_ NE-DNA Signal, Median (Q1–Q3)		0.51 (0.16–0.90)	0.26 (0.08–0.73)	0.65 (0.35–0.91)	*p* < 0.05
PCT, Median (Q1–Q3)		0.24 (0.08–2.32)	0.08 (0.06–0.11)	2.32 (0.46–13.56)	*p* < 0.01

## Data Availability

The data shown in this manuscript is available from the corresponding author upon request.

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
