# Peer review of "Biophysical Changes of Leukocyte Activation (and NETosis) in the Cellular Host Response to Sepsis"

_diagnostics, 2023, doi:10.3390/diagnostics13081435_

Round 1

Reviewer 1 Report

Reviewer Comments to authors

Regarding the study entitled "Biophysical changes of leukocyte activation (and NETosis) in the cellular host response to sepsis" with accept.

1-      The English used correct and readable.

2-      The work had a significant contribution to the field.

3-      The work was well organized and comprehensively described.

4-      There were appropriate and adequate references to related and previous work.

5-      Figure and Tables are selected properly and related to the main research question.

Author Response

We thank Reviewer 1 for recommending this article for publication in Diagnostics. 

Reviewer 2 Report

Dear Author (s)

1. There are several grammatical errors.

2. Please add the full name of each abbreviation for the first time. For example, PMA in abstract, ..., ISI, ...

3.  Please add ethical code with its details.

4. Who is "Our Lady"?

5. Please add the first letter of first name/surname of authors in Author Contributions.

6. Please add full name of each abbreviation below each figure or table.

7. Please take a look at the references. For example, you delete "Available: 342
https://journals.lww.com/ccejournal/Fulltext/2021/06000/Assessment_of_a_Cellular_Host_Response_Test_as_a.27.aspx" for reference 31. Or add the page for eference 31, ...

8. What is clinical significance?

9. What is novelty? We did see the previous studies (https://pubmed.ncbi.nlm.nih.gov/34151282/) that had a conclusion similar you.

10. Please add a discription from main outcomes in methods. What is ISI range?

11. You need reference for some sentence in methods. For example, after 10 minutes ...

Author Response

We thank Reviewer 2 for their recommended revisions to the article. We have addressed these revisions in the resubmitted text and have, point by point, addressed these in the cover letter for resubmission. We have uploaded the cover letter here as well as in the resubmission for Reviewer 2 to see how we have addressed these points. 

Reviewer 3 Report

Dear Authors,

this is a well-written manuscript, presenting the evaluation of IntelliSep test as a rapid in-vitro diagnostic tool of dysregulated immunity in sepsis and support ED physicians in timely diagnosis of sepsis. I have no questions or queries pertaining to your manuscript.

Best Regards

Author Response

We thank Reviewer 3 for recommending this article for publication in Diagnostics. 

Round 2

Reviewer 2 Report

Dear

The manuscript is suitable for publication.